# A Policy Gradient Algorithm to Alleviate the Multi-Agent Value Overestimation Problem in Complex Environments

**DOI:** 10.3390/s23239520

**Published:** 2023-11-30

**Authors:** Yang Yang, Jiang Li, Jinyong Hou, Ye Wang, Huadong Zhao

**Affiliations:** 1Changchun Institute of Optics, Fine Mechanics and Physics, Chinese Academy of Sciences, Changchun 130033, China; yangyang214@mails.ucas.ac.cn (Y.Y.);; 2University of Chinese Academy of Sciences, Beijing 100049, China; 3Unit 32802 of the Chinese People’s Liberation Army, Beijing 100191, China

**Keywords:** deep deterministic policy gradient, playback of experience, group decision-making, overestimation of value function

## Abstract

Multi-agent reinforcement learning excels at addressing group intelligent decision-making problems involving sequential decision-making. In particular, in complex, high-dimensional state and action spaces, it imposes higher demands on the reliability, stability, and adaptability of decision algorithms. The reinforcement learning algorithm based on the multi-agent deep strategy gradient incorporates a function approximation method using discriminant networks. However, this can lead to estimation errors when agents evaluate action values, thereby reducing model reliability and stability and resulting in challenging convergence. With the increasing complexity of the environment, there is a decline in the quality of experience collected by the experience playback pool, resulting in low efficiency of the sampling stage and difficulties in algorithm convergence. To address these challenges, we propose an innovative approach called the empirical clustering layer-based multi-agent dual dueling policy gradient (ECL-MAD3PG) algorithm. Experimental results demonstrate that our ECL-MAD3PG algorithm outperforms other methods in various complex environments, demonstrating a remarkable 9.1% improvement in mission completion compared to MADDPG within the context of complex UAV cooperative combat scenarios.

## 1. Introduction

With the emergence of intelligent algorithms, various industries are actively incorporating intelligent technologies to enhance efficiency and generate greater value. As artificial intelligence technology advances, there will be a means to connect individual agents with distinct functions, thereby augmenting system performance in terms of intelligence. In a multi-agent system, each agent possesses specific perceptions, interactions, and execution capabilities [1]. To collectively accomplish specific tasks, the multi-agent control method serves as a coordination approach that guides the behavior of each agent while considering inter-agent interactions and achieving common objectives. Recently, the utilization of multi-agent group control algorithms has become prevalent in domains such as cluster control [2] of unmanned aerial vehicles (uavs), distribution logistic path planning [3], distribution network optimization [4], and intelligent decision-making [5].

Currently, three types of decision support methodologies primarily exist; one is the expert system-based decision approach. An expert system refers to a computer software system developed within a specific domain by leveraging expert knowledge and experience [6]. The emulation of the cognitive process employed by human experts in a specific field enables commanders to receive more precise recommendations, thereby enhancing their decision-making efficiency. In an intelligent decision-making environment, expert system technology proves invaluable for optimizing command plans, improving situational awareness, and ultimately facilitating efficient decision-making. Sun, L. et al. introduced a joint decision system based on an expert system [7], which encompasses knowledge and rules from multiple fields and leverages the knowledge representation and reasoning capabilities of expert systems to support effective decision-making. The expert system possesses automatic characteristics [8] and requires minimal manual intervention. However, due to its limited self-learning ability, the expert system remains constrained by human-defined rules during operation, serving as a computer-based substitute for repetitive and rudimentary tasks. The second approach involves model-based swarm intelligent decision algorithms with numerous representative studies available. For instance, Ma, Y. et al. [9] employed an enhanced swarm algorithm in UAS teamwork defense scenarios that significantly improved task success rates while Hu, Z.Z. et al. [10], based on an optimized particle swarm algorithm, effectively obtained opponent target strategies and maximized their utilization. Li, J. et al. [11] successfully achieved an efficient optimal solution for the confrontation strategy by employing an enhanced ant colony algorithm. In terms of situational awareness during confrontations, a swarm intelligence algorithm is utilized to accurately identify and locate the opponent’s position and actions. As scientific and technological advancements continue to progress, intelligent decision-making scenarios necessitate the consideration of exponentially increasing factors, thereby demanding extensive data processing and analysis [12]. Consequently, researchers have recently been exploring a third approach that integrates AI technology into autonomous decision-making processes in order to address these challenges effectively. The third approach involves an autonomous decision-making method that leverages artificial intelligence technology. This innovative technique integrates machine learning algorithms [13] to enable self-learning and self-optimization, surpassing the limitations imposed by known rules and models. By effectively processing vast amounts of data, this methodology extracts valuable insights to make precise decisions even in complex environments. However, it is important to note that this method is currently at the research and experimental stage and has not yet reached full maturity or widespread adoption.

In recent years, deep reinforcement learning has become an important research topic in the field of artificial intelligence. Because it allows agents to extract relevant state information directly from the environment and due to its excellent perceptual exploration ability, it can optimally adapt to complex and dynamic environments, so it has been widely used in real-time strategies. Popular multi-agent reinforcement learning algorithms integrate value-learning and strategy-learning structures. It is worth noting that reference [14] uses the multi-agent depth strategy gradient algorithm to solve the challenges brought by the complex and uncertain dynamic environment encountered in the UAV cluster confrontation process. Reference [15] proposes a multi-agent cooperative combat simulation algorithm based on reinforcement learning to achieve a balanced decision-making method. Reference [16] introduces a valuable multi-agent reinforcement learning algorithm for training flight controllers in aircraft simulators to improve their performance in air confrontation. Reference [17] proposes a multi-agent reinforcement learning algorithm, combining Q-learning and attention mechanisms to solve path-planning problems. However, the mature reinforcement learning algorithms used so far in complex environments are mainly based on the integration of value-learning and strategy-learning methods, so this algorithm inherits some limitations of value-based reinforcement learning methods.

Based on the challenges encountered in current mature algorithm research and engineering practices, it has been identified that existing research exhibits the following limitations:(1)The complexity of simulation environments leads to an expanded observation space and action space for agents, resulting in issues such as subpar experience quality collected by the experience playback pool, inefficient sampling, and challenging algorithm convergence.(2)With an increasing number of agents and environmental complexities, classical algorithm evaluation modules require substantial computational resources and are susceptible to problems like imprecise evaluations, reduced adaptability of classical algorithms, low task completion rates, and arduous convergence.

Although the aforementioned challenges have been effectively addressed in single-agent reinforcement learning, there are limited approaches available to tackle these issues within the intricate multi-agent domain. DeepMind introduced the DQN algorithm in 2013 [18], which emphasized that the value function typically represents the state value function for assessing the agent’s action quality based on a specific strategy within the current state. In this study, deep learning neural network technology is employed to estimate the Q-value function, while experiential replay and target network technology are utilized to enhance the traditional Q-learning algorithm. Despite significant advancements achieved by the DQN algorithm, notable errors in value estimation still persist. The double DQN algorithm [19], proposed by Hado et al. in 2016, effectively mitigates the loss caused by overestimation inherent in the DQN algorithm and further enhances the performance of DQN. In 2018, Scott et al. introduced the TD3 algorithm [20], which successfully addresses the issue of Q overestimation in the DDPG algorithm and significantly stabilizes its training process within continuous action spaces. To enhance cooperative capabilities in multi-UAV aerial countermeasures, Zhang, D. et al. devised a collaborative maneuver decision method based on a multi-agent double delay depth deterministic strategy gradient [21].

In this paper, we propose the adversarial discriminant network based on the multi-agent depth strategy gradient to accurately and effectively analyze the relative contribution of the agent’s input state and action. To optimize the accuracy of value function estimation, we introduce two critical networks for estimating the action value function jointly. Additionally, we propose a preferential experience replay mechanism based on cluster stratification to enable agents to fully leverage experience information, thereby enhancing algorithmic learning efficiency and stability. Experimental results demonstrate that our improved algorithm exhibits superior convergence effects and task completion compared to other reinforcement learning algorithms.

## 2. Related Work

Reinforcement learning excels in handling sequential decision tasks, with its fundamental modeling tool being the Markov decision process (MDP). An MDP typically comprises the current state space, action space, reward function, and next state space. The state space (S) represents the set of all feasible states. The action space (A), on the other hand, denotes the collection of all possible actions. The reward function (R) quantifies the value that an environment provides to an agent upon performing a specific action. Ultimately, reinforcement learning aims to maximize this reward (u) by optimizing its expectations. The return can be calculated using Formula (1); in this formula, t represents the time step:(1)$$u_{t}=R_{t}+R_{t+1}+R_{t+2}+R_{t+3}+\ldots +R_{n}$$

In MDP, the discount return is usually used and is calculated as follows:(2)$$u_{t}=R_{t}+\gamma R_{t+1}+\gamma^{2}R_{t+2}+\gamma^{3}R_{t+3}+\ldots$$
In this formula, γ represents the discount rate.

In both single-agent and multi-agent cooperative and adversarial environments, the commonly employed and effective algorithms in reinforcement learning can be broadly categorized into three groups: value-based reinforcement learning algorithms, exemplified by DQN; policy gradient-based reinforcement learning algorithms, represented by DDPG; and actor–critic-based reinforcement learning algorithms, such as MADDPG.

### 2.1. Basic Algorithm of Value-Based Reinforcement Learning

#### DQN

The DQN algorithm represents an advancement of the Q-learning algorithm and exhibits proficiency in handling intricate discrete environment tasks. The estimation of the Q-value function is accomplished through a neural network capable of processing extensive high-dimensional information, while its parameters are continually optimized during training to gradually approximate the actual Q-value function. During the training process of the DQN algorithm, two crucial techniques are employed that have a profound impact on subsequent reinforcement learning algorithms: the experiential replay mechanism and target network.

The replay mechanism, commonly known as the replay buffer, serves multiple purposes in facilitating smooth training by reducing sample correlation, enhancing data utilization, and mitigating data imbalances. The size of the buffer is typically determined as a hyperparameter denoted by b. It should be noted that the buffer can only retain a maximum of b experiences, which are derived from distinct strategies independent of each other; consequently, when the buffer reaches its capacity, the oldest experiences are automatically removed.

The ‘Target’ function plays a crucial role in guiding the algorithm towards convergence to a stable state during the training process, thereby facilitating the acquisition of an improved strategy. When training the value network, it becomes essential to sample multiple batches of experience samples from the buffer and subsequently update the network parameters using a difference method, as described by Formula (3):(3)$$Q^{\pi }\left(s_{t},a_{t}\right)=r_{t}+Q^{\pi }\left(s_{t+1},\pi \left(s_{t+1}\right)\right)$$
Here, π represents the current policy. However, due to potential changes in TD targets, this can introduce instability into the training process. To address this issue, Figure 1 illustrates the objective function of DQN, which incorporates a Q network (Figure right) responsible for generating and fixing the target throughout training while only updating parameters on the left side. This configuration ensures that both target and Q networks remain fixed, enabling stable training.

### 2.2. Basic Algorithm of Strategy-Based Reinforcement Learning

#### 2.2.1. Deep Deterministic Policy Gradient

In the field of continuous control, the deep deterministic policy gradient (DDPG) is a renowned algorithm that utilizes neural networks to generate deterministic actions. DDPG extends the concept of DQN to accommodate continuous action spaces and shares similarities in training with DQN. However, unlike DQN, DDPG incorporates a policy network alongside the value network for action output and necessitates training both networks. The depicted structure in Figure 2 represents the architecture of the action-discriminant network. In this figure, state (s) serves as input for the policy network that outputs an action (a), while both the action (a) and state (s) are fed into the Q network as inputs, resulting in an output of action value.

The DDPG algorithm also adopts the target function method. As shown in the left part of Figure 3, there are four networks in the DDPG algorithm, namely the Q network, target Q network, policy network, and alongside target policy network. In the DDPG algorithm, the target Q network is responsible for generating the target Q-value, denoted as Target_Q, denoted as $T_{Q}$, which is calculated using Formula (4). In this formula, ω is the parameter of the network.
(4)$$T_{Q}=r+\gamma_{\bar{\omega }}\left(s^{\prime },a^{\prime }\right)$$

Target policy network, Target_A, noted as $T_{A}$. Calculate the next action $a^{\prime }$ using Formula (5):(5)$$T_{A}=a^{\prime }=\mu_{\bar{\theta }}\left(s^{\prime }\right)$$

The Q network calculates the current valuation using Formula (6):(6)$$Q=Q_{\omega }\left(s,a\right)$$

The policy network calculates the actions *a* to be taken in the current state using Formula (7):(7)$$a=\mu_{\theta }\left(s\right)$$

When training the DDPG algorithm, both loss functions should be optimized simultaneously. Equation (Equation 8) can calculate the loss function ($Loss_{1}$) of the target Q network and Q network, and the Q network is optimized by the mean-variance between the Q_target and Q_target.
(8)$$Loss_{1}=MSE\left[Q_{\omega }\left(s,a\right),r+\gamma Q_{\varpi }\left(s^{\prime },a^{\prime }\right)\right]$$

Equation (Equation 9) calculates the loss function ($Loss_{2}$) of the policy network:(9)$$Loss_{2}=-Q_{\omega }\left(s,a\right)$$

In Formula (9), $a=\mu_{\theta }\left(s\right)$ can be calculated using Formula (7).

#### 2.2.2. Multi-Agent Deep Deterministic Policy Gradient

The multi-agent deep deterministic policy gradient (MADDPG) algorithm is an extension of the DDPG algorithm designed specifically for multi-agent environments, incorporating adaptive improvements. This algorithm can be categorized as an actor–critic approach due to its utilization of both action and discriminant networks. As depicted in Figure 4, MADDPG employs individual action and discriminant networks for each agent, which bear resemblance to those used in DDPG.

Take $Agent_{i}$, for example: the agent has additional information in addition to its state action data, like actions $\left(a_{1},a_{2},a_{i},\ldots ,a_{N}\right)$ and states $\left(o_{1},o_{2},o_{i},\ldots ,o_{N}\right)$ of other agents. In addition, MADDPG also uses the empirical pooling mechanism to store the data $\left(s,a,r,s^{\prime }\right)$ generated by each agent interacting with the environment. Whenever new data are generated, these data are stored in the replay buffer, denoted as D. The training process takes the form of centralized training with decentralized execution, i.e., each agent, according to their strategy, obtains the current state of action and interacts with the environment to gain experience in their own experience buffer pool D. For all agents, after interacting with the environment, each agent randomly selects experiences from the pool to train their neural network. To accelerate the agent’s learning process, the critic network includes inputs of the other agents’ observation states and actions, updating the critic network parameters by minimizing the loss. Then, the parameters of the action network are calculated via the gradient descent method, finally realizing centralized training.

#### 2.2.3. Overvaluation Based on DDPG

This paper is based on the DDPG algorithm. In the DDPG algorithm, there are two main issues that need to be addressed. Firstly, during the training process, the estimation of the value function tends to be inaccurate, leading to instability in strategy updates and overestimation of Q-value functions. Secondly, essential noises exist in the target strategy which aids in exploring the state space but also contributes to high estimation errors in Q-value functions and affects estimator accuracy. Consequently, utilizing imprecise estimates in each update results in an accumulation of errors. This accumulated error can cause any unfavorable state to be erroneously estimated as favorable, ultimately resulting in suboptimal performance during strategy updates.

Scott et al., in their paper [20] (TD3), demonstrated that the action-discriminant approach updates the agent’s strategy based on an approximate estimation of the discriminant network, resulting in an overestimation of its value. Through a comparison between the actual and estimated Q-values obtained from different environments, they successfully confirmed the previous algorithm’s tendency to overvalue outcomes. Additionally, they provide Formula (10) for accurately calculating the true Q value: (10)$$\left.True_{Q}=\left[G_{t}|S_{t}=s\right]\right.$$
where $G_{t}$ is the cumulative return reward. Then the Formula (10) deformation is as follows: (11)$$\left.True_{Q}=\left[R_{t}+\gamma \left(R_{t+2}+\gamma R_{t+3}+\ldots \right)|S_{t}=s\right]\right.$$
where $\left(R_{t+2}+\gamma R_{t+3}+\ldots \right)$ is exactly $G_{t+1}$. Therefore, the true Q-value is calculated as follows: (12)$$\left.True_{Q}=\left[R_{t}+\gamma G_{t+1}|S_{t}=s\right]\right.$$

Based on the methodologies employed in previous studies, this paper adopts the MADDPG algorithm to evaluate the Q estimation within a highly authoritative experimental environment pertaining to multi-agent reinforcement learning, as depicted in Figure 5. In accordance with Scott et al.’s research, if the agent’s estimated Q-value surpasses the actual Q-value at any given step size under a uniform time step size, it signifies an overestimation of the algorithm’s Q-value. In the two figures depicted in Figure 5, the blue curve denotes the estimated Q-value while the red curve represents its actual counterpart. Notably, both curves exhibit a significant disparity with the blue curve surpassing the red one, indicating an evident overestimation of Q-values.

As shown in Figure 5a, the experiment was performed in the simple_tag environment. In Figure 5b, the experiment is shown in the simple_adversary environment.

## 3. Proposed Algorithm

The multi-agent reinforcement learning environment is characterized by instability, high dimensionality, and continuous space, which pose challenges for Q estimation in this complex setting. Multi-agent reinforcement learning algorithms often require updating numerous parameters and utilizing significant computing resources, leading to cumulative bias. Additionally, issues such as the low correlation between state and action and sparse experience can arise in the experience playback pool, making it difficult to determine sampling points that are uniform enough or sufficient in number to affect algorithmic efficacy. To address these problems, we propose the multi-agent dual dueling policy gradient algorithm based on the empirical clustering layer (ECL-MAD3PG) algorithm as an improvement over MADDPG with three key enhancements.

### 3.1. The Dueling-Critic Network

As depicted in Figure 6a, the input of the critic network in the MADDPG algorithm is compressed into multiple layers of parallel, fully connected networks. However, this structure fails to accurately and effectively analyze the relative contributions of each state and action. Additionally, due to the mixing of state value and action value, when processing high-dimensional state spaces, a large number of parameters need to be learned by the critic network to evaluate the action network’s strategy. Consequently, this leads to substantial computational costs and overestimation issues. To address these limitations and precisely assess the relative contribution of each action by the critical network, we propose introducing an advantage function. This involves dividing the critic network into two components: an advantage function and a value function, thereby forming an adversarial discrimination network structure. The specific splitting process is as follows:(1)Define the optimal advantage function

$Q^{*}\left(s,a\right)$ is defined as the optimal action value function, $V^{*}\left(s\right)$ is the optimal state value function, and their calculation formula is as follows:(13)$$\left\{\begin{array}{c} Q^{*}\left(s,a\right)=\max_{\pi }Q_{\pi }\left(s,a\right)\\ V^{*}\left(s\right)=\max_{\pi }V_{\pi }\left(s\right) \end{array}\right.$$

The formula for calculating the optimal advantage function is as follows:(14)$$A^{*}\left(s,a\right)=Q^{*}\left(s,a\right)-V^{*}\left(s\right)$$

In the context where $V^{*}\left(s\right)$ assesses the quality of the state(s) and $Q^{*}\left(s,a\right)$ evaluates the quality of the agent’s action within that state(s), Formula(20) can be considered as equivalent to utilizing $V^{*}\left(s\right)$ as a reference point, while $A^{*}\left(s,a\right)$ represents the superiority of action(a) compared to baseline $V^{*}\left(s\right)$.

(2)Properties of the Advantage function

In the field of reinforcement learning theory, function $V^{*}\left(s\right)$ maximizes function $Q^{*}\left(s,a\right)$ with respect to *a*, as given by Equation (Equation 15):(15)$$V^{*}\left(s\right)=\max_{a}Q^{*}\left(s,a\right)$$

Taking the maximum of the action *a* on both sides of Equation (Equation 14) results in Equation (Equation 16):(16)$$\max_{a}A^{*}\left(s,a\right)=\max_{a}Q^{*}\left(s,a\right)-V^{*}\left(s\right)$$

Substituting Equation (Equation 15) into Equation (Equation 16) yields Equation (Equation 17):(17)$$\max_{a}A^{*}\left(s,a\right)=0$$

(3)Dueling-Critic function

According to the definition of the optimal advantage function, i.e., Equation (Equation 14), it can be transformed into the following equation:(18)$$Q^{*}\left(s,a\right)=V^{*}\left(s\right)+A^{*}\left(s,a\right)$$

Substituting Equation (Equation 17) into Equation (Equation 18) yields the formula for the optimal value function, as shown in Equation (Equation 19):(19)$$Q^{*}\left(s,a\right)=V^{*}\left(s\right)+A^{*}\left(s,a\right)-\max_{a}A^{*}\left(s,a\right)$$

Equation (Equation 19) serves as the theoretical foundation for the modification and decomposition of the critic network. Figure 6b represents the schematic diagram, illustrating the improvement made to the original critic network structure in this paper through Equation (Equation 19):

Figure 6a illustrates the original critic network structure of the MADDPG algorithm, which comprises fully connected layers. Figure 6b presents the novel architecture for this problem, referred to as ‘dueling-critic’. In this configuration, the input of the critic network remains unchanged, consisting of state (s) and action (a). However, there is a modification in the transmission mode: both state (s) and action (a) vectors are initially gathered by a fully connected layer before being separately transmitted to the baseline value network V(s;θ). The data undergoes compression in the dominant network A(s,a;θ) and ultimately undergoes vector splicing through another layer of fully connected networks. In this structural framework, the critic network effectively maintains state value functions and action value functions, respectively. This dual-value function approach enhances the discriminant capability of the agent’s network during the learning process, thereby strengthening the correlation between state value and action value estimations. Consequently, it leads to more accurate Q-value estimation while also mitigating any imbalance that may exist between state and action values.

### 3.2. Priority Experience Replay Mechanism with Conditional Constraints

As the complexity of the experimental environment in the reinforcement learning algorithms increases, it leads to a rise in the number of agents within the system and an expansion in the dimensions of states and observations. This severely impacts the efficiency and quality of algorithm training. Wu Mingxi et al. [22] have proposed that traditional experience replay methods store all experiences obtained from agent–environment interactions, directly into the experience pool. During training, experiences are randomly sampled from this pool for learning. The original experience replay mechanism does not distinguish which experiences are more valuable. Storing every experience without discernment means that while high-quality experiences can be beneficial for further training of the algorithm, the sampling process may select a large number of low-quality experiences. This leads to a reduced training efficiency, consuming significant time. PER (proposed by Yang, S. et al. [23]) alleviates the aforementioned problems. The basic idea behind PER is just to adopt one replay buffer for the sake of conserving experiences, assigning a priority level to each one, replacing uniform sampling with non-uniform sampling. In this mechanism, the priority level is represented by the absolute value of the temporal difference error, i.e., $\left|\delta_{j}\right|$. A larger $\left|\delta_{j}\right|$ indicates that the algorithm’s evaluation of the state action value at that moment is inaccurate, so that experience should be given a higher weight. During sampling, there are two methods to compute the sampling probability. The first method calculates the probability using Formula (20): (20)$$\left.p_{j}\propto \left|\delta_{j}\right|+\varepsilon \right.$$

In this formula, ε is defined as a small number to inhibit the sampling possibility from reaching 0, ensuring that the overall specimens are plotted via a non-zero probability. The second sampling method sorts $\left|\delta_{j}\right|$ in descending order, and then calculates the sampling probability using Formula (21):(21)$$p_{j}\propto \frac{1}{rank(j)}$$

In this expression, rank(j) is the index of $\left|\delta_{j}\right|$, and the larger $\left|\delta_{j}\right|$ is, the smaller rank(j) becomes. The principles behind the two sampling methods mentioned above are consistent: the larger $\left|\delta_{j}\right|$ is, the higher the probability of the sample being selected. Because this is non-uniform sampling, different samples have different sampling probabilities, leading to a bias in the algorithm’s prediction. At this point, the learning rate should be adjusted to offset the bias caused by different sampling probabilities. The structure of the priority experience replay is presented in Table 1.

In Table 1, *b* represents the array size, which must be manually set. If the number of samples exceeds *b*, the oldest samples in the replay pool need to be removed. Using this approach, as the algorithm learns, it can sequentially select experiences for training from the replay buffer in order of priority, from highest to lowest, thereby maximizing the utilization of experience information and enhancing both learning efficiency and stability. This grants the algorithm several advantages: it retains essential experiences in the replay buffer, ensuring that these experiences still have a high priority under the current policy, and can be learned first, preventing the propagation of erroneous information. In traditional experience replay processes, since the sampled experiences are entirely random, some critical experiences may not be selected, leading to learning outcomes that fall short of expectations. In contrast, priority experience replay maintains the diversity of the experience pool by setting priorities, ensuring that experiences with greater variance and representativeness are reused more frequently, thereby enhancing the stability of learning. However, when multi-agent reinforcement learning algorithms are applied to relatively complex real-world tasks, they face intricate state and observation dimensions. This places a high demand on the efficiency of the algorithm’s training. Although the priority experience replay mechanism can prioritize the extraction of high-quality experiences, it suffers from high time complexity. As a result, in practice, the algorithm efficacy improvement is often not significant, and it heavily consumes the system’s computational power.

The conventional method of experience playback is limited by its random sampling approach, which occasionally results in the omission of crucial experiences and even yields subpar-quality recovered experiences. Consequently, this hampers the anticipated learning outcomes. In order to enhance the quality of experiences stored in the experience playback pool, it is imperative to investigate the recycling mechanism and experience sampling mechanism that governs this pool. Building upon the original experience replay pool depicted in Figure 7a, this paper proposes a preferential replay mechanism based on experience clustering and stratification. This is achieved by incorporating a condition module prior to the retrieval of experiences from the original experience pool, as illustrated in Figure 7b. The purpose of the condition module is to effectively cluster the array of experiences, wherein batches of experiences are grouped based on their priority and subsequently evaluated for their eigenvalues. These eigenvalues are then stored within the experience pool in descending order, ensuring that experiences with higher values take precedence over those with lower values. In this manner, during the interactive learning phase with the environment, the algorithm can effectively prioritize experiences from the condition module in a hierarchical order based on their significance. Additionally, by employing clustering techniques, it can further establish hierarchical priorities to fully leverage experience information and enhance both learning efficiency and stability.

The specific workflow of the priority playback mechanism based on empirical clustering stratification is as follows:

Step 1: The agent interacts with the environment to obtain a collection of empirical data, and the algorithm assigns priority to the data based on the magnitude of the TD error it exhibits. This prioritized list of data is represented as $\left(s_{j},a_{j},r_{j},s_{j+1},\delta_{j}\right)$.

Step 2: Cache multiple sets of experience data lists in the ‘Condition’ module and conduct clustering operations to group experiences and calculate the lambda value for each group.

Step 3: According to the characteristic value ($\delta_{j}$), the experiential data are stored in the playback pool (buffer) in ascending order of magnitude.

Step 4: In the sampling stage, experience is selectively sampled based on its value($\delta_{j}$), enabling the agent to prioritize experiences from groups with higher eigenvalues.

Step 5: The modified algorithm employs empirical data to enhance the optimization of network parameters.

## 4. Environment Setup and Experimental Analysis

In this study, we employ experimental environments with varying levels of environmental complexity to validate the progressive and adaptable nature of our ECL-MAD3PG algorithm. Specifically, we evaluate the performance of our algorithm by comparing it against PerMAD3, MADDPG, and MATD3 algorithms in MPE. Additionally, we independently develop a multi-agent UAV cooperative countermeasure platform to assess the superiority and reliability of our ECL-MAD3PG algorithm when compared to PerMAD3, MADDPG, and MATD3 algorithms. The experiments are conducted on a Core i7-11700K processor running a Win10 system. Our algorithm is partially implemented using Python 3.8 programming language along with PyTorch deep learning framework and third-party libraries such as NumPy and Pygame.

### 4.1. Multi-Agent Particle Environment

In this paper, we focus on the multi-particle environment (MPE), a two-dimensional, time-discrete, and spatially continuous multi-agent environment developed by OpenAI. MPE is specifically designed to simulate and verify a range of multi-agent reinforcement learning algorithms. It achieves this by controlling the movements of various particles within a two-dimensional space, enabling the accomplishment of diverse tasks. To provide a comprehensive experimental analysis, we compare MPE with two well-established and widely used classical multi-agent environments. Figure 8a indicates the simple_tag environment, while Figure 8b indicates the simple_adversary environment.

#### 4.1.1. Simple_Tag Environment

Figure 8a depicts the Simple_Tag environment from the official examples, also known as a predator–prey environment. As described in the official documentation, this environment contains four agents and two black obstacles, the quantity of which can be customized. Typically, the green prey agents move faster, and they receive a negative reward if hit by the red adversary agents. The three predators move slower and must employ effective strategies to successfully capture the prey while avoiding obstacles. The game ends for the round if the prey is captured or if any agent goes out of bounds. By default, there is one prey, three predators, and two obstacles.

(1)Action Space Design

Both the prey and predators share the same action space, as outlined in Table 2. The action space has a dimension of 5, with the following specific actions: move up, move down, left, right, and stop.

(2)State Space Design

Both the prey and predators have largely identical state spaces, as detailed in Table 3. The state space for the agents has a dimension of 7.

(3)Reward Function Design

Superior green prey agents move faster and receive a negative reward when hit by the red adversary agents (a penalty of −10 for each collision). Adversary agents, which move slower, are rewarded when they successfully collide with the superior agent (+10 for each collision). When predators and prey are within a certain range, both predators and prey receive rewards. The game round ends if the prey is captured or if any agent goes out of bounds.

(4)Key parameters of the ECL-MAD3PG algorithm

After conducting multiple iterations, a comprehensive set of parameters that exhibit optimal performance for the ECL-MAD3PG algorithm in this experimental setting have been documented and are presented in Table 4.

#### 4.1.2. Simple_Adversary Environment

Figure 8b depicts the Simple_Adversary Environment from the official examples. In this environment, there is one red opposing agent, N actor agents, and N landmarks, with N being the default set to 2. In this task, all agents can observe the sites of landmarks and other agents. One of the landmarks is the ‘destination landmark’, which is highly crucial. Only the actor agents know about it, and their goal is to coordinate with each other to deceive the red agent and prevent it from reaching the ‘destination landmark’. The game round ends if an agent goes out of bounds.

(1)Action Space Design

The action space is consistent for both the opposing agent and actor agents, as outlined in Table 5. The action space has a dimension of 5, with specific actions being move up, move down, left, right, and stop.

(2)State Space Design

The state space for the opposing agent and actor agent is essentially the same. The state space for actor agents is provided in Table 6, with a state space dimension of 7.

(3)Reward Function Design

The actor agents cooperate with each other, and if any one is close enough to the target location, all actor agents receive the same reward. The opposing agent can also receive a reward if it is close enough to the target location, but it needs to guess which one is the target.

(4)Key parameters of the ECL-MAD3PG algorithm

After conducting multiple iterations, a comprehensive set of parameters that exhibit optimal performance for the ECL-MAD3PG algorithm in this experimental setting have been documented and are presented in Table 7.

### 4.2. The Multi-UAV Cooperative Combat Simulation Platform

In this study, a sophisticated cooperative countermeasure platform (MACCSP) involving multiple unmanned aerial vehicles is devised to validate the robustness and consistency of the algorithm. In this particular environment, the primary objectives of multi-UAV cooperation encompass utilizing a diverse range of sensors carried by the UAVs to thoroughly explore the protected objects belonging to the blue unit. During this exploration process, it becomes imperative to locate and interfere with the radar system employed by the blue square unit in order to ascertain a relatively secure flight path for the pilot positioned behind the drone cluster. The four illustrations depicted in Figure 9 aptly demonstrate and simulate the entire scene’s progression.

Figure 9a depicts the initial state of the scene, wherein red units are positioned on the left and blue units on the right. Figure 9b illustrates the exploration simulation process undertaken by a UAV cluster within the red unit. Subsequently, in Figure 9c, we observe the UAV cluster exploring both red and blue unit radar systems while simultaneously interfering with them. This interference leads to a reduction in their detection radius. Finally, as depicted in Figure 9d, upon completion of exploration and jamming tasks, the UAV cluster successfully acquires a relatively secure flight path to evade radar detection.

#### 4.2.1. Establishing the Red Team Model

The probability of the red team’s UAVs detecting the radar signals from the blue team:(22)$$P=1-\exp\left\{-C\int_{t_{0}}^{t_{1}}{\left[{\left(x_{t}-\varsigma \right)}^{2}+{\left(y_{t}-\xi \right)}^{2}\right]}^{-2}dt\right\}$$

Formula (22) indicates the probability of detecting the blue protection target within time $\left(t_{0},t_{1}\right)$, where ς represents the target’s horizontal coordinate and ξ represents the target’s vertical coordinate.

#### 4.2.2. Establishing the Blue Team Model

When the target enters the radar’s detection range, it is detected by the radar with a certain probability determined by the energy interaction between them. For the purpose of this study, we exclusively examine and analyze both slow scanning and fast scanning techniques employed in radar systems.

**(1)** Slow Radar Scans

During slow radar scans, the radar’s detection of the target can be considered as discrete observations. In this case, the radar’s detection probability $P_{D}$ is represented as follows:(23)$$P_{D}=1-\prod_{i=1}^{m}\left(1-P_{dt}\right)$$

In Formula (23), ‘m’ represents the number of radar object contacts over the continuous period, and it can be computed using Formula (24)
(24)$$m=\left[\frac{t}{t_{search}}\right]$$

$t_{search}$ represents the radar’s cycle. In Formula (23), $P_{dt}$ denotes the probability of detecting the target during the i-th contact. In the absence of electronic interference, this can be calculated using Formula (25)
(25)$$P_{dt}={\left[\frac{n_{0}S_{N_{t}}+1}{n_{0}S_{N_{t}}}\right]}^{n-1}\exp\left\{-\frac{Y_{0}}{n_{0}S_{N_{t}}+1}\right\}$$

**(2)** Rapid Radar Scanning

During rapid radar scanning, it can be regarded as a continuous observation. In the absence of interference, the probability of the radar detecting a point target is as follows:(26)$$P=1-\exp\left\{-C\int_{0}^{t}R{\left(t\right)}^{-4}dt\right\}$$

Let $U=C\int_{0}^{t}R{\left(t\right)}^{-4}dt$ denote the detection potential. Up to time t, the detection potential of the radar to the target detection zone is as follows:(27)$$U=C\int_{0}^{t}R{\left(t\right)}^{-4}dt=C{\int_{0}^{t}\left[X_{0}^{2}+{\left(Y_{0}+V_{\delta }t\right)}^{2}\right]}^{-2}dt$$

Let $y_{0}+V_{\delta }t=X_{0}\tan\varphi$, then:(28)$$U\left(X_{0}\right)=\frac{C}{V_{\delta }X_{0}^{3}}\int_{\phi_{0}}^{\phi }\cos^{2}\phi d\phi =\frac{C}{2V_{\delta }X_{0}^{3}}\left[\left(\phi -\phi_{0}\right)+0.5\left(\sin2\phi -\sin2\phi_{0}\right)\right]$$

The probability of detecting the target in this segment is as follows:(29)$$P_{X_{0}}=1-e^{-U(x_{0})}$$

The probability of radar blocking the red drone:(30)$$P=1-\exp\left\{\frac{\alpha W_{d}^{\beta }}{\sigma_{d}^{\beta }}\right\}$$
Here, α, β, and σ are proportional parameters.

#### 4.2.3. Action Space Design

With the aim of reducing the dimensionality of the UAV swarm’s action space, this paper introduced a combination of discretization and simplification for certain actions. The actions of the agents are presented in Table 8, having a dimension of 8.

Table 8 specifies the interference frequency bands for the UAV as follows: low frequency band (0.03 GHz–1 GHz), medium frequency band (1 GHz–15 GHz), and high-frequency band (15 GHz–30 GHz). By combining actions based on the aforementioned table, an action space vector of length 3780 can be generated. All action choices are encoded in the one-hot encoding format.

#### 4.2.4. State Space Design

As for the UAV, its state space comprises two sections: the first section is the circumstance state space *S*, representing the whole circumstance state; the second part is the agent’s observation *O*, representing the UAV’s own state, as well as its observations of the environment. Table 9 is the global state space and Table 10 is the local state space.

#### 4.2.5. Reward Function Design

Considering that UAVs need to cooperate to complete the task, if the distance between them is too far, communication will be impossible. Therefore, a distance reward $R_{1}$ between the UAVs should be set up.
(31)$$R_{1}=\left\{\begin{array}{cc} 100\times (\frac{C}{D}-1) & D>C\\ 0 & D\leq C \end{array}\right\}$$

In the formula, *C* represents the communication distance between UAVs; *D* represents the actual distance between UAVs. The reward when approaching the target is given by:(32)$$R_{2}=\left\{\begin{array}{cc} 500\times \frac{D_{blue}-d_{now}}{d_{last}-D_{red}} & d_{now}>D_{blue}-2\\ 100\times (1-\frac{d_{now}}{D_{blue}}) & other \end{array}\right\}$$

Reward $R_{3}$ for being detected by the radar:(33)$$R_{3}=-10$$

Reward $R_{4}$ for detecting a radar:(34)$$R_{4}=20$$

Reward $R_{5}$ for interfering with the radar:(35)$$R_{5}=1000\times \left(1-\frac{D_{blue\_now}}{D_{blue}}\right)$$

In the formula, $D_{blue\_now}$ denotes the detection distance of the radar after interference, and $D_{blue}$ represents the maximum detection distance of the radar.

Reward $R_{6}$ for the drone blocked:(36)$$R_{6}=-100$$

Reward $R_{7}$ for opening up the pilot paths:(37)$$R_{7}=100$$

Total reward *R:*(38)$$R=R_{1}+R_{2}+R_{3}+R_{4}+R_{5}+R_{6}+R_{7}$$

#### 4.2.6. Design of the Task Completion Degree

According to the simulation task, it can be devised to ascertain the successful targeting of the blue square position in each round. In case of a successful attack, $D_{i}\;=1$; otherwise, $D_{i}\;=0$. The formula for evaluating the level of task accomplishment is as follows:

The main task of this scene is as follows: The red drone fleet flies in front to detect and interfere with the radar of the mine, opening up a suitable flight route for the pilots flying behind to reach the blue protection unit. In each round, we observe whether a path is generated. If there is a path generation, it is set to $D_{i}\;=1$, otherwise $D_{i}\;=0$. The formula for calculating the degree of task completion is as follows:(39)$$\sigma_{D}=\frac{n}{N}$$
where: *n* represents the number of rounds in which $D_{i}\;=1$, and *N* denotes the total number of rounds.

#### 4.2.7. Key Hyperparameters of the PerMAD3 Algorithm

After conducting multiple iterations, a comprehensive set of parameters that exhibit optimal performance for the ECL-MAD3PG algorithm in this experimental setting have been documented and are presented in Table 11.

### 4.3. Experimental Analysis in Multi-Agent Particle Environments

The ultimate objective of reinforcement learning is to acquire the optimal strategy by means of agent–environment interactions, aiming to maximize cumulative return. Hence, a crucial metric for comparing the superiority of reinforcement learning algorithms lies in their average reward returns. In this study, we employ the multi-agent particle swarm environment (MPE) as our primary experimental platform and train four algorithms within two multi-agent environments: simple_tag and simple_adversary on the MPE platform. Figure 10 presents two graphs illustrating the average reward curves of these four algorithms across the aforementioned multi-agent environments. The x-axis represents the number of training rounds undergone by each algorithm, while the y-axis denotes their corresponding average reward values.

In the experimental setting of simple_tag, we sequentially execute four algorithms, namely ECL-MAD3PG, PerMAD3, MADDPG, and MATD3. The resulting average reward values are depicted in Figure 10a; as illustrated in the figure, all four algorithms undergo an equal number of training rounds (8,000,000 times), and eventually converge to a stable state. The four algorithms are currently in the initial learning phase, spanning from round 0 to the 310,000th round. During this stage, the agent effectively optimizes neural network parameters by continuously engaging with the environment and acquiring high-quality empirical data through recovery and sampling. A comparative analysis between our algorithm and Per-MAD3 at this stage reveals that our average reward curve surpasses that of Per-MAD3 algorithm both in terms of value and rate of improvement. This observation indicates that our proposed cluster-based hierarchical priority playback mechanism outperforms the original PER scheme in this specific environment. Similarly, when comparing our algorithm with MATD3, we find that our proposed dual adversarial discriminant network exhibits superior performance compared to the dual value network scheme (MATD3) within this particular environment.

In the experimental environment of simple_adversary, we sequentially execute the ECL-MAD3PG, PerMAD3, MADDPG, and MATD3 algorithms and plot the average reward value in Figure 10b. As depicted in the figure, all four algorithms are trained for an equal number of rounds (3,000,000 times), and each method demonstrates a tendency toward stability after training. The four algorithms are currently in the initial learning phase, ranging from round 0 to 800,000. During this stage, the agent can enhance the parameters of the neural network through continuous interaction with the environment, enabling recovery and sampling of high-quality empirical data. By comparing our algorithm with PerMAD3 at this stage, it becomes evident that our average reward curve surpasses that of PerMAD3 algorithm both in terms of value and rate of improvement. This observation indicates that our proposed cluster-based hierarchical priority playback mechanism outperforms the original PER scheme in this particular environment. Similarly, when comparing our algorithm with the MATD3 algorithm, we find that our proposed dual adversarial discriminant network exhibits superior performance compared to the dual value network scheme (MATD3) within this environment.

As illustrated in Figure 10, the four algorithms underwent distinct training rounds in two separate environments and eventually converged to a stable return value after a specific number of iterations. By analyzing the fluctuation pattern of the return value curve depicted in Figure 10a, this study computed the average reward values for each algorithm within rounds 300,000 to 8,000,000, which were subsequently recorded in Table 12. Similarly, based on the observed changes in the return value curve exemplified in Figure 10b, this research calculated and documented the average reward values for all four algorithms from round 500,000 to round 3,000,000 in Table 13.

Based on the values presented in Table 12, it can be inferred that the algorithm proposed in this study (ECL-MAD3PG) exhibits a superior reward value compared to alternative algorithms, thereby indicating its enhanced performance within this specific environment.

Moreover, based on the values presented in Table 13, it can be inferred that the algorithm proposed in this study (ECL-MAD3PG) exhibits a superior reward value compared to alternative algorithms, thereby indicating its enhanced performance within this specific environment.

In addition, experiments are conducted to enhance the accuracy of Q-value estimation. In this study, the actual Q-value of the agent in each state is computed using Formula (12), and both the real value and predicted value obtained from the algorithm’s value network are recorded and stored separately. By plotting the Q-value curve, a comparative analysis is performed to evaluate the algorithm’s estimation capability. Figure 11 illustrates two graphs depicting Q estimation before and after implementing algorithmic improvements. The degree to which the problem of Q overestimation is mitigated is typically assessed using metrics such as the mean absolute error or root-mean-square error. These metrics are employed to measure the disparity between the currently estimated Q-value and the true Q-value. A close-to-zero value of this error indicator indicates that the algorithm effectively reduces Q overestimation. Based on experimental data collected in two different environments, Figure 11 presents two groups of results for calculating the root-mean-square error between the current estimated Q-values and actual Q-values using both algorithms. The corresponding outcomes are summarized in Table 14.

According to the statistical results presented in Table 14, in both experimental environments, the ECL-MAD3PG algorithm proposed in this paper demonstrates the root-mean-square errors (RMSEs) between the estimated Q-values and the true Q-values that are closer to 0 compared to the MADDPG algorithm. This indicates that the ECL-MAD3PG algorithm possesses the capability to mitigate the issue of Q-value overestimation.

In conclusion, the comparisons of four algorithms in two different environments demonstrate that the ECL-MAD3PG algorithm not only improves the average return performance but also mitigates the issue of Q-value overestimation, commonly associated with policy gradient reinforcement learning algorithms. Consequently, this algorithm exhibits strong convergence and reliability.

### 4.4. Analysis of Experiments in Electronic Warfare Environments

To validate the adaptability and superiority of our algorithm, we developed a sophisticated multi-agent reinforcement learning experimental environment called the multi-UAV cooperative combat simulation platform (MACCSP). Within this platform, four algorithms, namely ECL-MAD3PG, PerMAD3, MADDPG, and MATD3, were individually trained for 2 million rounds. The reward values obtained from each round were meticulously recorded and saved. In order to evaluate the algorithm’s performance in complex tasks within a multi-agent reinforcement learning environment, two key metrics were employed: the average reward value and the degree of completion of collaborative tasks by the UAV cluster. Figure 12 presents both experimental results and statistical findings to support our claims.

Figure 12a illustrates that in the MACCSP environment of comparable complexity, the average return of all four algorithm groups tends to stabilize after approximately 250,000 rounds. The attainment of convergence, post-simulation, indicates a reasonable platform design. Table 15 records the average reward value for each algorithm group following the aforementioned round count, revealing superior performance by the ECL-MAD3PG algorithm relative to its counterparts. In addition, Figure 12b demonstrates that the ECL-MAD3PG algorithm outperforms other algorithms in terms of achieving the highest level of task completion in the UAV cluster coordination. In summary, the ECL-MAD3PG algorithm exhibits notable advantages and remarkable adaptability within complex environments.

In addition, Figure 12b demonstrates that the ECL-MAD3PG algorithm outperforms other algorithms in terms of achieving the highest level of task completion in the UAV cluster coordination. In summary, the ECL-MAD3PG algorithm exhibits notable advantages and remarkable adaptability within complex environments.

## 5. Discussion

In this study, we compare three algorithms: ECL-MAD3PG (our proposed algorithm), PER-MADDPG, MADDPG, and MATD3, in three different environments. We calculate the average reward value for each algorithm in these environments and plot a curve to illustrate the trend of change in these values. As our work builds upon MADDPG, the algorithm is still influenced by certain hyperparameters, which may require multiple adjustments to achieve convergence.

Notably, we find that the learning rate of the discriminant network significantly impacts training effectiveness. However, other parameters such as the policy network learning rate, experience replay pool size, and noise can be set to commonly used values without significant impact on results. Taking the “simple_tag” environment in the MPE platform as an example, Figure 13 presents the experimental results of two groups of discriminant network learning rates using the ECL-MAD3PG algorithm. Comparing these results with those depicted by the ECL-MAD3PG curve in Figure 10a, it can be inferred that while the average reward value of the algorithm increases with an augmented discriminant network learning rate, convergence becomes challenging. Subsequent experiments demonstrate that within this environment, a learning rate ranging from $1\times 10^{-4}$ to $2\times 10^{-4}$ yields relatively favorable outcomes. Beyond this range, excessively large learning rates lead to heightened volatility in algorithmic results and even failure to converge. The default values for parameters such as the policy network’s learning rate, experience playback pool size, and noise exhibit minimal influence on experimental outcomes.

By employing operations akin to fine-tuning the learning rate of the discriminant network, this study successfully determined the optimal experimental parameters through a series of rigorous experiments conducted in three distinct environments. The obtained results are meticulously documented and are presented in Table 16 for reference.

After conducting multiple experiments in three different environments, the obtained results were documented in this table. The findings indicate that ECL-MAD3PG exhibits superior performance in relatively simple environments when employing a smaller discriminant network learning rate. Conversely, in more complex environments, ECL-MAD3PG demonstrates enhanced efficacy with the utilization of a larger discriminant network learning rate.

## 6. Conclusions

In this paper, we propose the ECL-MAD3PG algorithm, which addresses the issue of estimation error in the agent action value evaluation caused by function approximation of its internal discriminant network, based on the multi-agent depth strategy gradient algorithm. This leads to reduced reliability and stability of the model, resulting in convergence difficulties. With the increasing complexity of the environment, there is a decrease in the quality of experience collected by the experience playback pool, leading to inefficiency in the sampling stage and further hindering algorithm convergence. To overcome these challenges, our proposed ECL-MAD3PG algorithm incorporates a multi-agent dual confrontation policy gradient approach, along with empirical clustering stratification.

(1)This paper proposes a dual adversarial critic network structure to accurately and effectively analyze the relative contribution of the input state and action of the agent and estimate the optimal value network, so as to optimize the algorithm and its critic component(2)This paper proposes a preferential experience playback based on experience clustering, which makes full use of the experience information obtained by the interaction between the agent and the environment, improves the quality of the recovered experience, and thus improves the learning efficiency and stability of the algorithm.(3)In this paper, we independently design a complex multi-UAV cooperative confrontation environment, and verify the adaptability and stability of the algorithm, and the task completion rate of the algorithm is improved by 9.1% compared with other classical reinforcement learning algorithms.

## Figures and Tables

**Figure 1 sensors-23-09520-f001:**
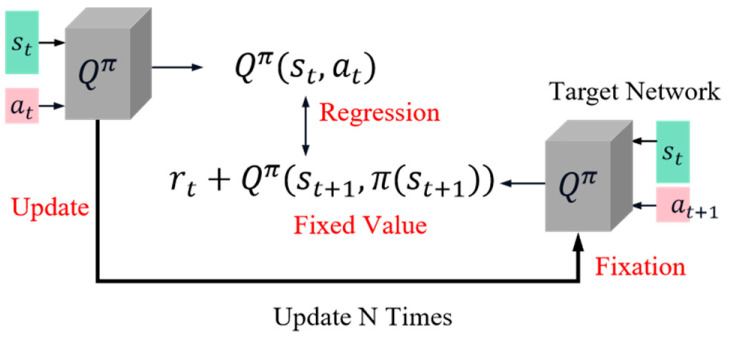
The target network in DQN.

**Figure 2 sensors-23-09520-f002:**
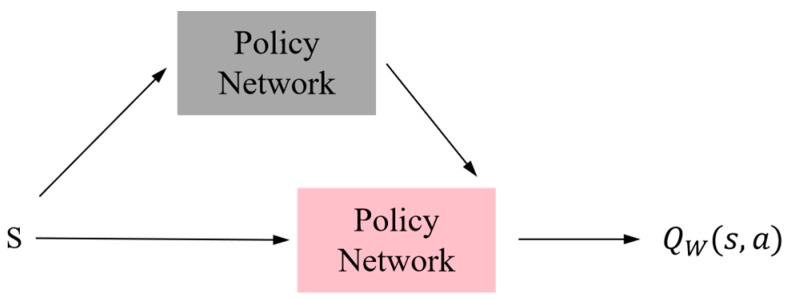
Actor–critic structure.

**Figure 3 sensors-23-09520-f003:**
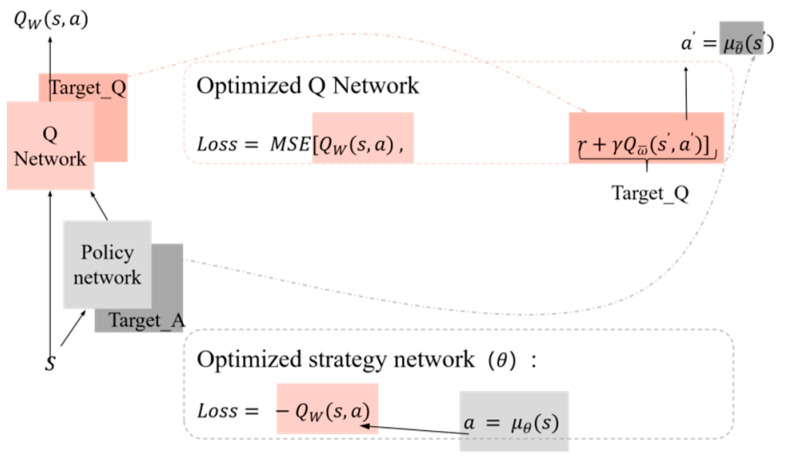
The A-C structure of DDPG and the construction of loss function.

**Figure 4 sensors-23-09520-f004:**
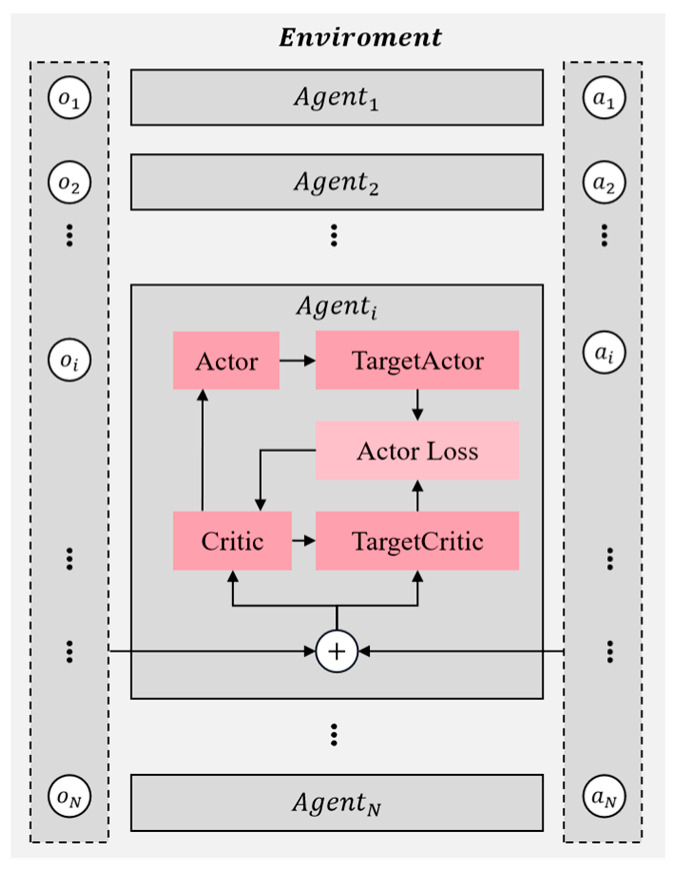
Each agent in the MADDPG algorithm has an actor–critic structure.

**Figure 5 sensors-23-09520-f005:**
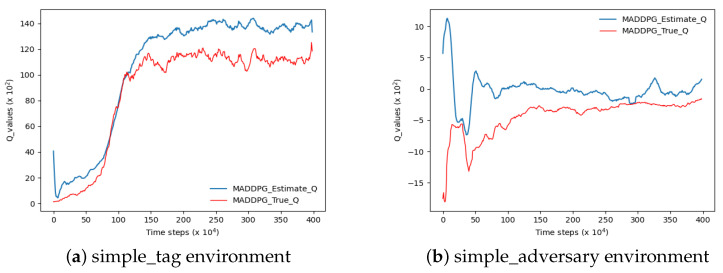
The overestimation of the Q-value of MADDPG in two environments. The blue curve represents the estimated value and the red curve represents the true value.

**Figure 6 sensors-23-09520-f006:**
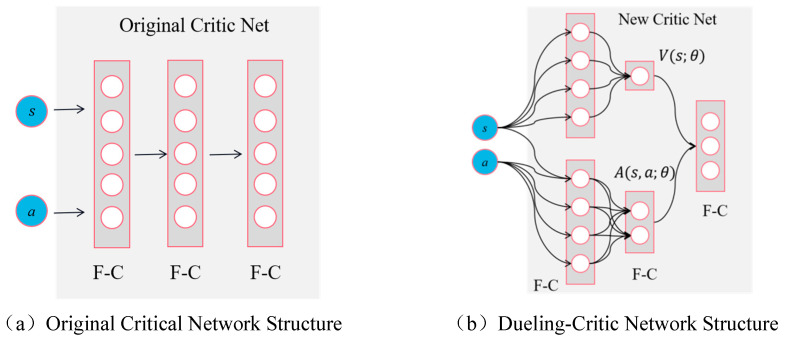
The critic network structure in MADDPG has improved. From left to right: (**a**) shows the structure of the original discriminant network, where “F-C” represents a fully connected network; (**b**) shows the dueling-critic network structure proposed in this paper.

**Figure 7 sensors-23-09520-f007:**
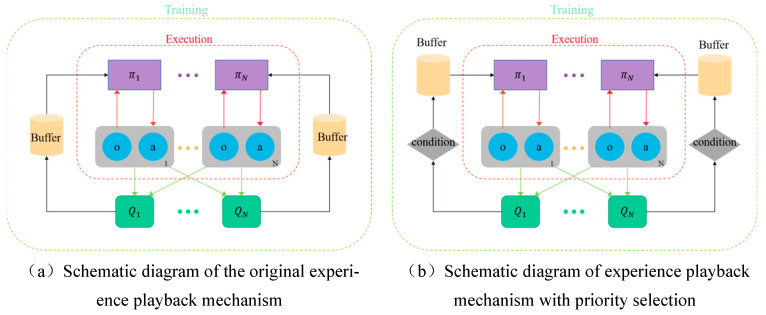
Comparison of experience playback mechanism before and after improvement. From left to right: (**a**) shows the original experience playback mechanism, (**b**) is a conditional module-based priority experience playback mechanism.

**Figure 8 sensors-23-09520-f008:**
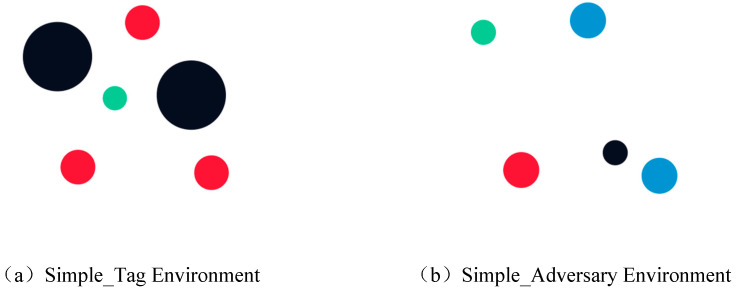
Two multi-agent environments in the MPE platform. From left to right: the larger two black circles in figure (**a**) represent obstacles, where the smaller circle represents the agent; The two circles of the same size in figure (**b**) represent landmarks, and the other three circles of the same size represent agents.

**Figure 9 sensors-23-09520-f009:**
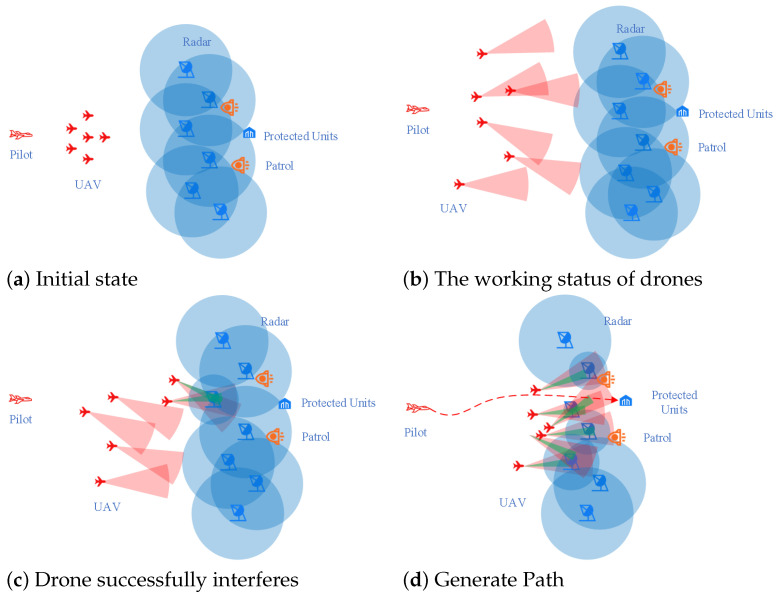
Experimental process in MACCSP.

**Figure 10 sensors-23-09520-f010:**
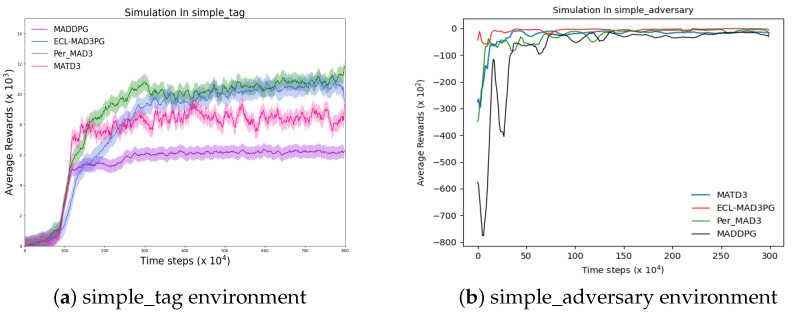
Comparison curve of real the Q-value and estimated Q-value before and after algorithm improvement. The algorithm proposed in this paper is ECL-MAD3PG.

**Figure 11 sensors-23-09520-f011:**
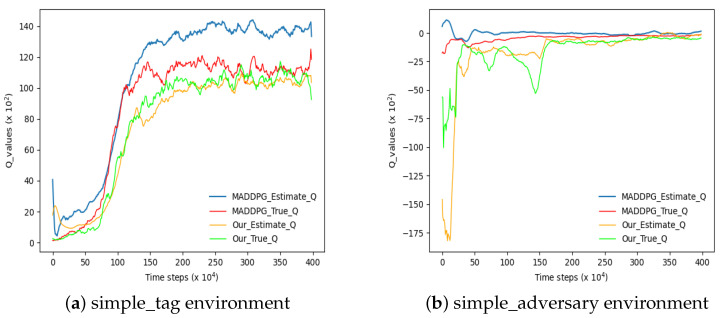
Comparison curve of the real Q-value and estimated Q-value before and after algorithm improvement. The “Our” in this figure represents our proposed algorithm “ECL-MAD3PG”.

**Figure 12 sensors-23-09520-f012:**
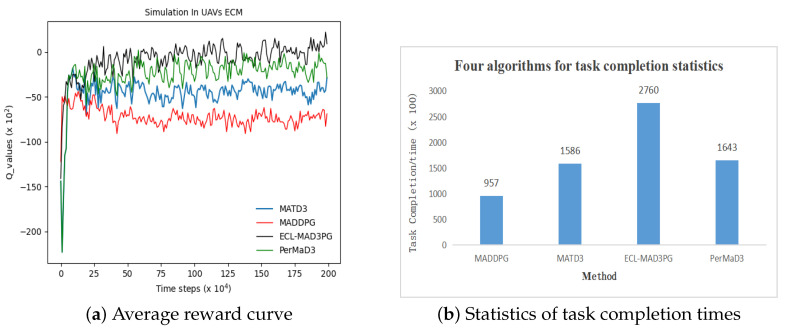
Experimental results of four algorithms in MACCSP.

**Figure 13 sensors-23-09520-f013:**
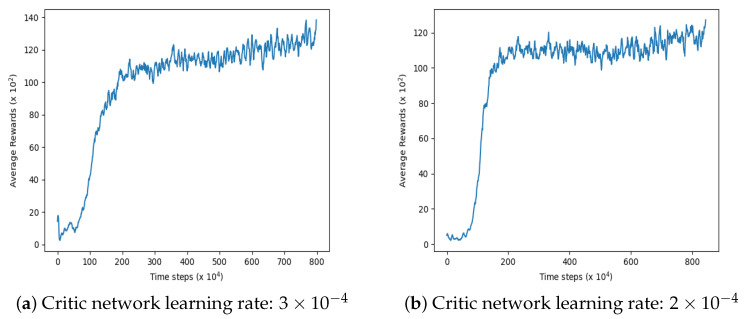
In the “simple_tag” environment, two groups of different discriminating network learning rate experiment. The algorithm is using ECL-MAD3PG.

**Table 1 sensors-23-09520-t001:** Priority experience playback array.

Number	Experience List	TD Error	$p_{j}$	Learning Rate
....	....	....	....	....
j−1	$\left(s_{j-1},a_{j-1},r_{j-1},s_{j}\right)$	$\delta_{j-1}$	$\left.p_{j}\propto \left|\delta_{j-1}\right|+\varepsilon \right.$	$\alpha \cdot {\left(b\cdot p_{j-1}\right)}^{-\beta }$
*j*	$\left(s_{j},a_{j},r_{j},s_{j+1}\right)$	$\delta_{j}$	$\left.p_{j}\propto \left|\delta_{j}\right|+\varepsilon \right.$	$\alpha \cdot {\left(b\cdot p_{}\right)}^{-\beta }$
j+1	$\left(s_{j+1},a_{j+1},r_{j+1},s_{j+2}\right)$	$\delta_{j+1}$	$\left.p_{j}\propto \left|\delta_{j+1}\right|+\varepsilon \right.$	$\alpha \cdot {\left(b\cdot p_{j+1}\right)}^{-\beta }$
....	....	....	....	....

**Table 2 sensors-23-09520-t002:** Action state table.

Field	Description
move_up	Up
move_down	Down
move_left	Left
move_right	Right
Stop	Stop

**Table 3 sensors-23-09520-t003:** Observation table.

Field	Description
Self_vel	Self Velocity
Self_pos	Self Position
landmark_rel_pos	Position of obstacles relative to oneself
other_agent_rel_pos	Position of same-type agents relative to oneself
Opp_agent_rel_pos	Position of opposing agents relative to oneself
Other_agent_rel_vel	Velocity of same-type agents relative to oneself
Opp_agent_rel_vel	Velocity of opposing agents relative to oneself

**Table 4 sensors-23-09520-t004:** Simple_tag hyperparameter table.

Hyperparameter	Value
Actor Learning Rate	$1\times 10^{-3}$
Critic Learning Rate	$1\times 10^{-4}$
Buffer Size	$3\times 10^{5}$
Batch Size	64
Noise	0.1
Discount rate	0.85

**Table 5 sensors-23-09520-t005:** Action state table.

Field	Description
Move up	move_up
Move down	move_down
Left	move_left
Right	move_right
Stop	Stop

**Table 6 sensors-23-09520-t006:** Observation table.

Field	Description
Self_vel	Self Velocity
Self_pos	Self Position
Goal_rel_pos	Position of the goal
Other_landmark_pos	Position of the other landmark
other_agent_rel_pos	Position of same-type agents relative to oneself
Opp_agent_rel_pos	Position of opposing agents relative to oneself
other_agent_rel_vel	Velocity of same-type agents relative to oneself
Opp_agent_rel_vel	Velocity of opposing agents relative to oneself

**Table 7 sensors-23-09520-t007:** Simple_adversary hyperparameter table.

Hyperparameter	Value
Actor Learning Rate	$1\times 10^{-3}$
Critic Learning Rate	$2\times 10^{-4}$
Buffer Size	$3\times 10^{5}$
Batch Size	64
Noise	0.1
Discount rate	0.9

**Table 8 sensors-23-09520-t008:** Action state.

Field	Description
Flying Actions	Forward, Backward, Left, Right, Hover
Flying Speed	Low, Medium, High
Reconnaissance Direction	Left Front, Direct Front, Right Front
Interference Intensity	0, Low, Medium, High
Interference Frequency Band	Low, Medium, High
Interference Target	Consistent with the number of radars.

**Table 9 sensors-23-09520-t009:** Environment state table.

Field	Description
Target Location	The location of the blue-protected unit
Radar Location	Coordinates of the ground radar
Radar Frequency	Frequency of the ground radar’s signal emission
Radar Detection Range	Real-time detection distance of the radar

**Table 10 sensors-23-09520-t010:** Agent observation table.

Field	Description
Position	UAV coordinates
Orientation	UAV flight direction
Speed	UAV flight speed
Direction	UAV’s targeted reconnaissance direction
Intensity	UAV’s interference intensity
Frequency band	UAV’s interference frequency band
Flight Time	UAV’s remaining life value
Flag	Is interference activated
Target	Located radar position

**Table 11 sensors-23-09520-t011:** UAV electronic countermeasure hyperparameter table.

Hyperparameter	Value
Actor Learning Rate	$5\times 10^{-3}$
Critic Learning Rate	$5\times 10^{-4}$
Buffer Size	$5\times 10^{5}$
Batch Size	256
Noise	0.1
Discount rate	0.9

**Table 12 sensors-23-09520-t012:** The average reward value from 3,000,000 to 8,000,000.

Algorithm	Average Reward
ECL-MAD3PG	1047.166
PerMAD3	1009.641
MATD3	845.795
MADDPG	616.031

**Table 13 sensors-23-09520-t013:** The average reward value from 500,000 to 3,000,000.

Algorithm	Average Reward
ECL-MAD3PG	−543.053
PerMAD3	−2370.871
MATD3	−1654.441
MADDPG	−3071.483

**Table 14 sensors-23-09520-t014:** Comparison of the root-mean-square errors between the predicted Q-values and the real Q-values for the four methods.

Simple_Tag Environment		Simple_Adversary Environment	
MADDPG	ECL-MAD3PG	MADDPG	ECL-MAD3PG
25.956	9.021	4.154	2.742

**Table 15 sensors-23-09520-t015:** The average reward value of rounds, from the 500,000th to the 2,000,000th.

Algorithm	Average Reward
ECL-MAD3PG	−1
PerMAD3	−38
MATD3	−39
MADDPG	−66

**Table 16 sensors-23-09520-t016:** The relatively stable parameters of ECL-MAD3PG in three environments.

Environment	Critic Network Learning Rate	Policy Network Learning Rate	Buffer Size	Noise
simple_tag	$1\times 10^{-4}$	$1\times 10^{-3}$	$3\times 10^{5}$	0.1
simple_adversary	$2\times 10^{-4}$	$1\times 10^{-3}$	$3\times 10^{5}$	0.1
MACCSP	$5\times 10^{-4}$	$5\times 10^{-3}$	$5\times 10^{5}$	0.1

## Data Availability

Data are contained within the article.

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
