# Peer review of "A Policy Gradient Algorithm to Alleviate the Multi-Agent Value Overestimation Problem in Complex Environments"

_sensors, 2023, doi:10.3390/s23239520_

Round 1
Reviewer 1 Report
Comments and Suggestions for Authors
This paper proposes a Dueling-Critic network based on the original Critic network structure for multi-agent systems. Moreover, simulation results show the effectiveness of the proposed network. The language of this paper needs to thoroughly modified. There are some grammar errors such as Line 107, Line 256, and so on. In addition, some figures, such as Figure 5, Figure 7, etc, are not clear such that the readers cannot estimate the performance of the proposed network structure. In a word, there are several contributions in this paper.
Comments on the Quality of English LanguageThe language of this paper needs to thoroughly modified. There are some grammar errors such as Line 107, Line 256, and so on.
Author Response
Thanks to the reviewer, please check the attachment.

Reviewer 2 Report
Comments and Suggestions for Authors
The paper is well written and structured.
The scientific contribution is sufficient.
1. Clearly describe the practical applications that can be deployed, based on this solutions.
2. In the Discussion, include a table, that compares the results and outcomes of the proposed study with the existing related studies.
Comments on the Quality of English Language
Proofread the paper
Author Response

(The authors gave the same response as above.)

Reviewer 3 Report
Comments and Suggestions for Authors
In this paper, an improved method of MADDPG algorithm is proposed, from both theoretical and experimental aspects, it is innovative to some extent, but there are some places that need to be further improved, such as:
1) Since MATD3 is an improved version of MADDPG algorithm, why not directly improve this method on MATD3? As can be seen from the curve in Figure 10, when the training times are enough, MATD3 is close to the improved version in this paper;
2) The content of related work in Section 2 can be simplified, which is common sense for people in the industry.
3) Whether the Q value is overestimated or not should be explained by the conclusions given in the existing literature, related theories or popular experience in principle, the curve given by the example in Figure 5 is not convincing enough.
4) The most important contribution of this paper is that Dueling-Critic network added to MADDPG, but in the experimental part, no any information to demonstrate how it works. The author should also introduce how the conditional module in Figure 7b works in detail in the experimental part, and how to ensure the validity of the condition module.
5) It is suggested that a set of empirical parameter which are reasonable and convenient for readers to refer should be given in the discussion part, the general explanation do not has much reference value. For CondMaD3 algorithm, whether there are some reference intervals for a set of optimal configuration parameter values?
If “ In different environments, the hyperparameter settings for the CondMaD3 algorithm also differed significantly.”, does that mean the algorithm performs better than other algorithms just depend on a specific environment or a different set of parameter?
Author Response

(The authors gave the same response as above.)

Reviewer 4 Report
Comments and Suggestions for Authors
This paper presents a study on improving multi-agent reinforcement learning. However, the paper is poorly presented and is difficult to follow. A major revision is required.
All abbreviations should be defined when they first appear. For example, UAV in page 2 is not defined and the authors need to check throughout the paper for other cases.
All variables used in equations should be defined when they first appear. For example, in Eq(1), what are t and n? What is \gama in Eq(2)? The authors need to check all other equations to make sure that any undefined variables should be defined when they first appear. Also equations need to be discussed in the text.
Some concepts or terminologies should be defined. For example, what is “value function”?
In the sentence bellow Eq(12), the authors mention “this chapter using …, the test results are shown in Figure 5, …”. By this chapter, do you mean your current paper or other people’s paper? You need to add reference if talking other people’s paper. What are the results in Figure 5 and what problem or case study is it? Also the sentence is very long and poorly presented.
In the experimental result section, the authors compare their proposed method with 4 other methods. However, Figure 10 and Tables 12 and 13 only show the performance of the 4 other methods. What about your proposed method?
Comments on the Quality of English LanguageEnglish is poor with many grammatical errors.
Author Response

(The authors gave the same response as above.)

Round 2
Reviewer 1 Report
Comments and Suggestions for Authors
This paper has been revised according to the comments.
Reviewer 3 Report
Comments and Suggestions for Authors
The author answered and revised all the questions.
Reviewer 4 Report
Comments and Suggestions for Authors
The authors have adequately addressed my comments and the revised manuscript can be accepted.